# Awareness and Availability of Low Sodium Iodized Salt: Results from Formative Research of Promoting Uptake of Low SodiUm Iodized Salt by Rural and Urban HousehoLds in India—The PLURAL Study

**DOI:** 10.3390/nu16010130

**Published:** 2023-12-30

**Authors:** Reena Sehgal, Nikhil Srinivasapura Venkateshmurthy, Rajesh Khatkar, Shiva Prasad Konkati, Prashant Jarhyan, Manika Sharma, Nicole Ide, Dorairaj Prabhakaran, Sailesh Mohan

**Affiliations:** 1Public Health Foundation of India, New Delhi 110030, India; reena.sehgal@phfi.org (R.S.); rajesh@phfi.org (R.K.); prashant.jarhyan@phfi.org (P.J.); dprabhakaran@phfi.org (D.P.); smohan@phfi.org (S.M.); 2Centre for Chronic Disease Control, New Delhi 110016, India; shiva@ccdcindia.org; 3Resolve to Save Lives, New York, NY 10004, USA; msharma@resolvetosavelives.org (M.S.); nide@resolvetosavelives.org (N.I.); 4Department of Non-communicable Disease Epidemiology, London School of Hygiene and Tropical Medicine, London WC1E 7HT, UK; 5School of Nursing & Midwifery, Deakin University, Burwood, VIC 3125, Australia

**Keywords:** sodium, hypertension, low sodium salt substitute, India

## Abstract

Dietary sodium intake is high among adults in India. Use of low sodium iodized salt (LSIS) can help reduce sodium intake. However, contextually relevant and culturally appropriate interventions to promote LSIS uptake in India have not been developed and implemented. We carried out formative research to inform an intervention to promote uptake of LSIS among rural and urban households in north (Sonipat district) and south (Visakhapatnam and Anakapalli districts) India. Sixty-two in-depth interviews of six focus groups were held with a range of stakeholders—consumers, retailers and influencers. Participant views on availability, affordability, taste and safety of LSIS, along with views on hypertension, its risk factors and potential intervention design and delivery strategies were elicited. Thematic analysis of the data was carried out. While the awareness of hypertension and its risk factors was high among the participants, awareness of LSIS was low. There was also low demand for, and availability of, LSIS. Since cost of LSIS was higher than regular salt, participants preferred that a subsidy be provided to offset the cost. Based on these findings, an intervention to promote the uptake of LSIS was implemented by project staff using various educational materials such as posters, pamphlets and short videos.

## 1. Introduction

Hypertension is the leading risk factor for cardiovascular disease, which in turn is the leading cause of mortality globally, as well as in India [1,2]. One-in-three adults in India are reported to have hypertension [3,4,5]. The high prevalence is in contrast with a low control rate of 12.6% [6,7]. One of the most important dietary risk factors for the prevention and control of hypertension is high dietary salt intake. While guidelines suggest a maximum daily dietary salt (sodium) intake of 5 (2) g/day/person, [8] data show that the average salt (sodium) intake is 8 (3.2) g/day/person among adult Indians [9].

The WHO Global Action Plan for the Prevention and Control of Non-Communicable Diseases (NCDs) as well as the Indian multi-sectoral action plan for NCDs recommends salt reduction as one of the key strategies to reduce population blood pressure and associated cardiovascular diseases [8,10]. The WHO global goal for reducing premature NCDs by 25% by 2025 has salt reduction as one of its key targets. The 25 by 25 goal recommends that countries reduce population salt intake by 30% by 2025 [8]. India has only one national measure in the form of a media campaign to reduce salt intake, earning it a score of two out of four according to the Global Report on Sodium Intake Reduction [11]. Although evidence exists on the impact of policies to reduce salt in processed and out of home foods [12,13], its impact on sodium intake in India will likely be limited given that most of the food consumed is made within the home. The salt intake from salt added while cooking or at the table together constituted 87.7% and 83.5% of the overall salt intake among adults in north and south India, respectively. The rest came from dietary sources such as meat, poultry and eggs, dairy and processed foods [14]. However, there have been recent increases in consumption of processed foods, especially in urban areas [15,16]. Hence, strategies to reduce salt intake at the household level should be the cornerstone of salt reduction efforts in the Indian context, alongside complementary measures to prevent increasing consumption of processed foods.

One of the effective, scalable and sustainable strategies to reduce salt consumption is to replace conventional salt with low sodium salt [17,18]. In the latter, typically 10–40% of sodium chloride is replaced with other minerals, most commonly potassium chloride [19]. Evidence from cluster randomised trials conducted in Peru [20] and China [21] has shown that low sodium salt use is associated with reduction of blood pressure among those with uncontrolled hypertension, reduced incidence of hypertension, stroke and major adverse cardiovascular events and lower risk of death. Moreover, low sodium salt use was found to be safe and acceptable.

While the results are encouraging, the intervention to promote low sodium iodized salt (LSIS) in India needs to be contextualized. Hence, we aim to explore issues around awareness, availability, affordability, taste and safety of LSIS [22,23] to design and implement an effective intervention to promote LSIS use by households in rural and urban India.

## 2. Materials and Methods

The present study was carried out as a part of the larger study entitled “**P**romoting Uptake of **L**ow Sodi**U**m Iodized Salt by **R**ural **A**nd Urban Househo**L**ds in India—The **PLURAL** Study”. Phase 1 of the PLURAL study consisted of formative research to develop contextually relevant interventions to promote the use of LSIS by households in Sonipat (north India) and Visakhapatnam and Anakapalli (south India) districts. Phase 2 of PLURAL was implementation of the intervention, whereas Phase 3 was its evaluation. The methods of formative research are described below. The PLURAL study was approved by the Institutional Ethics Committee of Public Health Foundation of India (approval number: TRC-IEC 465/21) on 1 July 2021.

### 2.1. Study Design

Formative research used qualitative research methods comprising face-to-face in-depth interviews and focus group discussions for data collection. We deployed qualitative methods as the goal was to attain an in-depth, contextual understanding of the issues around the topics of hypertension and its risk factors; relationship between dietary salt and hypertension; salt purchase and consumption patterns; LSIS availability, affordability, taste and safety.

### 2.2. Study Participants

The study population consisted of three distinct groups of participants—salt consumers, retailers and influencers. The consumers included women and men aged ≥ 30 years. Retailers were those selling salt in their shops and those supplying it. Influencers involved public healthcare staff (doctors, auxiliary nurse midwives, accredited social health activists), members of women’s self-help groups, sarpanches (heads) of the villages and members of resident welfare associations. Consumers provided insights on hypertension related issues, salt use, LSIS perception, factors influencing potential transition from regular salt to LSIS and exposure to health interventions. Retailers, in addition to all this, added a supply chain and logistical perspective about LSIS to the study. Influencers, given their vast experiences with the local community, gave an overview regarding health practices and perceptions, salt use and health interventions, along with approaches to LSIS introduction among the study population.

### 2.3. Participant Recruitment

We leveraged existing community partnerships developed over the last 10 years while implementing other research projects [24,25] in the study areas in order to purposively select our study participants. Project personnel initially met community stakeholders and gate keepers to explain the PLURAL study, and sought permission to recruit study participants to conduct interviews and focus group discussions. Using the information from the list of participants whom we had engaged with previously, the study personnel initially visited the potential participants and explained the PLURAL study to them. They were given a participant information sheet and consent form to read, and the opportunity to ask any questions they might have regarding the study or their participation. They were given a week’s time to decide on participation. We had initially planned to conduct a total of 40 in-depth interviews and four focus group discussions with the participants, but the final number of interviews and discussions was guided by data saturation.

### 2.4. Data Collection

After conducting a literature review, relevant topic areas were selected to be explored via in-depth interviews and focus group discussions with study participants. Our topic guide was also informed by a published framework [23] for addressing challenges to scale up low sodium salt use. The interviews allowed for the individual experiences and perspectives of participants to emerge regarding the topics of blood pressure, hypertension, perceptions about LSIS, salt reduction and transition to LSIS, whereas focus group discussions allowed us to explore group experiences or perceptions on health interventions, community perceptions and cultural norms, beliefs and relevant attributes of the population of interest with respect to hypertension, salt use and LSIS introduction in the community. We ensured the homogeneity of the participants in focus groups by conducting separate discussions with women and men. Each focus group had 8–10 participants belonging to the same socio-economic status. Thus, through interviews and focus group discussions, we were able to elicit individual and community perspectives, respectively. Topics within four broad categories, including awareness, affordability, taste and safety were explored with participants during interviews and discussions, which are shown in Table 1.

The interviews and focus groups were carried out by RS and SPK. After obtaining consent, the project personnel scheduled the interview and discussion at a date and time mutually convenient to the participants and the interviewers. Only the participants and the interviewers were present during the interview. A topic guide (See Appendix A) was used by the interviewers. The length of the interview varied between 30 to 45 min, whereas the focus groups lasted nearly 60 min. The interviews and focus groups were audio recorded and notes were taken.

### 2.5. Data Analysis

The audio recordings of interviews and focus group discussions were transcribed verbatim and translated into English from Hindi (language spoken in Sonipat) and Telugu (language spoken in Visakhapatnam and Anakapalli). The transcripts were uploaded to NVivo (Version 12, Lumivero, Denver, CO, USA) for analysis. A thematic analysis approach was used, wherein initial codes and a codebook were developed by RS. The codebook was used to code the data and organize the issues, ideas and topics within the data. Each issue/idea/topic was described and compared by participant type (consumer, retailer or influencer), residence (rural or urban) and site (Sonipat and Visakhapatnam/Anakapalli). Consistency in coding was maintained by continually referring back to the transcripts and the codebook. The coding was reviewed by NSV. These codes were clubbed into themes and supported by the addition of relevant quotations from the interview and discussion transcripts.

## 3. Results

A total of 62 and 60 participants were recruited for the in-depth interviews and focus groups, respectively. The mean, minimum and maximum ages of the participants were 47, 30 and 75 years, respectively. The majority of the participants (62%) were males. A total of 62 in-depth interviews and six focus groups were held across Sonipat and Visakhapatnam/Anakapalli sites. The detailed breakup of these numbers by site and type of participants is given in Table 2.

The themes, along with the description and verbatim quotes, are presented below. A summary of the findings by participant type is presented in Table 3.

### 3.1. Hypertension: Symptoms and Its Relationship with Diet and Lifestyle

All of the participants in this study were able to list some of the symptoms and/or risk factors of hypertension. Participants across both the study sites listed unhealthy diet (high in salt, sugar and fat), lack of physical activity, tobacco and alcohol use as some of the important risk factors for hypertension. Many were able to describe some of the symptoms (i.e., headache and dizziness) associated with hypertension. They were of the opinion that uncontrolled hypertension may lead to damage to vital organs like the heart, brain, kidneys and eyes, leading to other serious medical health conditions.


*“They say that high BP causes heart problems and paralysis stroke.”*
(Female, 42, Visakhapatnam; urban, Consumer)

The association between salt consumption and hypertension was known by almost all of the participants.


*“I have reduced salt and chili powder, I am not eating outside food, I have fruits and eat only home cooked food.”*
(Female, 45, Visakhapatnam; rural, Consumer)


*“I have heard that eating more salt can cause BP.”*
(Male, 41, Visakhapatnam; rural, Retailer)

The majority of participants believed diet and lifestyle changes, specifically the consumption of healthier food and an increase in physical activity, are important in keeping blood pressure under control.


*“Yes, so like going for morning walks or avoid eating fast food or eat fruits, it (hypertension) can be better.”*
(Male, 57, Sonipat; rural, Consumer)

### 3.2. Opinion, Experiences and Attitude towards Salt Reduction and Alternatives

According to many participants, taste was the main factor which hindered the reduction of salt in their diet or transitioning from regular salt to a healthier salt alternative.


*“I don’t like bland food or food with less salt.”*
(Female, 38, Sonipat; rural, Consumer)


*“The food doesn’t taste good if we reduce salt very much, there should be enough taste in the food.”*
(Female, 60, Visakhapatnam; urban, Consumer)

As taste is an important consideration for members within the household, participants opined that it was not feasible for food with low salt to be cooked separately for certain members of the family who are at risk of hypertension or have been advised by health professionals to reduce their salt intake.


*“If in family only one or two people have hypertension then it becomes difficult to cook separate food for them, that’s the biggest challenge.”*
(Male, 35, Sonipat; urban, Health Worker)

Health workers reported difficulty among patients to initiate a low sodium diet and adhere to it long-term.


*“They will stop eating salt for few days and they will also take medicines regularly but after few days when they start feeling normal again, they will start with the regular routine and again start eating everything because they will think they are alright now.”*
(Male, 35, Sonipat; urban, Health Worker)


*“When they come here for monthly follow up, we can tell them to take less salt and ghee. We tell them about diet and all. But they give preference to their taste. They don’t like the food without salt. So, they use extra salt while having food. We give them health education every time. We tell them to use less salt in their diet.”*
(Female, 35–50 group, Sonipat; rural, Community Health Officer)


*“We tell them to avoid pickles but they don’t do it. They don’t even restrict salt intake. Because they get used to that taste, it is difficult for them to stop it.”*
(Female, 31, Visakhapatnam; rural, Doctor)

### 3.3. Awareness of Low Sodium Iodized Salt

The study population across study sites had low awareness of LSIS. Most stakeholders, which included healthcare providers, were unaware of LSIS. Consumers became aware of LSIS for the first time during their participation in this study.


*“No, this is the first time I am hearing about it.”*
(Female, 60, Visakhapatnam; urban, Consumer)


*“Whatever we have been consuming or seen till date, we don’t know whether it’s low sodium or high sodium salt.”*
(Male, 45–60 group, Sonipat; rural, Consumer)


*“No, we haven’t heard about it. We got to know it from you just now and nowhere else.”*
(Male, 51, Sonipat; urban, Retailer)

### 3.4. Low Demand and Availability of Low Sodium Iodized Salt

The availability of LSIS was almost non-existent in both sites. While there was knowledge of LSIS as a product among a few retailers across urban sites in Visakhapatnam and Sonipat, the availability of LSIS was low due to lack of demand among urban consumers. LSIS awareness across rural sites was low among retailers. As there was no demand of LSIS by consumers, procurement of LSIS by retailers from salt distributors was not common.


*“It is available but only when there are sales, we can ask for availability, when there are no sales, we don’t keep it available.”*
(Female, 34, Visakhapatnam; urban, Retailer)


*“See, if any customers come, if they ask then I need to keep it compulsorily.”*
(Male, 38, Sonipat; urban, Retailer)


*“If it is not in the market then how will you create awareness, if any new company product comes in market, you create awareness that you use this in place of that, then only they will come to know, if you don’t have any idea, how will people get aware about it.”*
(Male, 45, Sonipat; urban, Retailer)

### 3.5. Affordability of Low Sodium Iodized Salt

The cost of LSIS was a significant factor in the purchase decision of consumers. The cost of LSIS was comparatively higher compared to regular salt. All stakeholders participating in this study were asked their opinions about the high cost of LSIS as a potential discouraging factor. The high price of LSIS was stated as a barrier for its purchase. 


*“Fewer people would prefer buying it, because of high price.”*
(Male, 45, Visakhapatnam; rural, Consumer)


*“No, here people will not buy something which is expensive, even if 5 rupees more then they will not buy, they will stop using it, but they will not buy.”*
(Male, 50, Sonipat; rural, Consumer)


*“No, if someone asks us then only, we get it as its rate is high.”*
(Female, 32, Visakhapatnam; urban, Retailer)

### 3.6. Opinion on Subsidy

Participants agreed that a subsidy would be important to offset the extra cost of LSIS. They emphasized the need for a subsidy, citing the economic burden that participants would be under when switching from regular salt to LSIS. 


*“If rate is more and it’s good for health then probably ten to twenty percent people will use in the beginning. And if you are giving the subsidy also and its good for health too, and we get to know about it, I think it would be eighty percent helpful.”*
(Male, 40–60 group, Sonipat; rural, Consumer)


*“If you give subsidy, it will help people. Price of many items has increased so if you give subsidy people will buy LSIS……If we get LSIS at cheaper price then we will be able to sell them”*
(Male, 50–60, Sonipat, Retailer)

### 3.7. Trusted Sources and Mode of Delivery of Health Messages

Across the study sites, healthcare professionals were mentioned as the most trusted source of information to promote the intake of LSIS. 


*“Through doctors also because doctors are next to God. Many people follow whatever a doctor says.”*
(Female, 31–50 group, Visakhapatnam; urban, Consumer)


*“These patients would always listen to the doctor. If the doctor prescribes them to use this alternative as it will be good for them, so they would 100% buy this. They will not buy it on our advice, we can only motivate. Like if a doctor prescribes a medicine, so we would take the medicine whether it is for hypertension or sugar. So, along with that if they will also mention about the salt so, people would buy it.”*
(Female, 31, Visakhapatnam; urban, Auxiliary Nurse Midwife)

In the case of rural areas in both study sites, the importance of the involvement of community leaders at the village level was also stressed by participants.


*“We should take their help also, otherwise people do not listen to us. They don’t listen if we go to them directly and tell them. If the sarpanch comes and tells, they will listen. Because they are village leaders they come and listen.”*
(Female, 33, Visakhapatnam; rural, Mid-Level Health Provider)

Participants believed the campaign should be interactive and engaging on an individual level. They preferred interpersonal communication delivered by trained project staff using educational materials like posters and pamphlets.


*“Yes, interpersonal communication will be good. Like ASHA workers, they will listen to them.”*
(Female, 35–35 group, Sonipat; rural, Doctor)


*“Door to door promotion by giving pamphlets and motivating them, we should have more interactions with them. I think nothing is better than face to face communication.”*
(Female, 31, Visakhapatnam; urban, Doctor)


*“During dengue season, the municipality sends people with mike to inform people about not letting water stand near their homes. Such type of communication if it goes to the people then they will buy LSIS”*
(Male, 42, Visakhapatnam; urban, Retailer)

## 4. Discussion

The aim of the study was to explore issues of awareness, availability, affordability, taste and safety of LSIS to inform development of an intervention to promote its uptake. The current study shows that while participants knew salt reduction is important for hypertension prevention and control, they were unaware that LSIS can help reduce salt consumption. The awareness on LSIS was low among participants across the study sites. Though urban retailers knew about LSIS as a product, they reported low demand as the reason for not selling/stocking it. High cost of LSIS was reported as a barrier to its use. Participants favoured provision of a subsidy to offset the cost of LSIS. Health professionals were reported as a trusted source for health-related and LSIS information. They preferred interpersonal communication by trained project staff using materials like posters and pamphlets as an intervention strategy to promote the uptake of LSIS. They also suggested involvement of community leaders in the intervention implementation.

All of the participants reported at least some of the risk factors for hypertension, its symptoms, consequences of uncontrolled hypertension and lifestyle changes to either prevent or control hypertension. This is explained by the fact that the investigators of the PLURAL study have been implementing research projects on hypertension, diabetes and food systems with active participation of community members and the local health system in the study site for the past 10 years [24,25]. Though participants were aware of salt reduction as a strategy to control blood pressure, they were unaware that use of LSIS can reduce blood pressure. Both awareness and availability of LSIS were low across the study sites and were influenced by one another. Awareness was low because LSIS was not available, whereas availability was low because consumers were not aware, and thus there was no demand. Studies from Peru [20], China [21] and Bangladesh [26] reported provision of LSIS free of cost to the study participants, and thus did not have to generate demand. At the time of conducting this study, the LSIS was priced 50% more than the regular salt. Given the high cost, participants preferred a subsidy to reduce the cost. Evidence shows that subsidies can promote increased purchase and consumption of healthy food [27]. Since the awareness of LSIS was low among participants, they did not report any concerns around its safety.

The results of this study have informed the design and implementation of the PLURAL study. Since the awareness of LSIS was low, the intervention focused on increasing it and thereby the demand for LSIS. Cost was reported as a major barrier for the use of LSIS. Hence, a subsidy was provided to offset the increased cost. Healthcare providers were trusted as the source for health-related information, so the educational materials included a fictional female doctor providing information on LSIS. The intervention was implemented by project staff through pamphlets, posters and a short video (90 s). The pamphlets and videos mainly targeted consumers, to be used during an interpersonal communication session during household visits. The posters were displayed mainly at shops and establishments to nudge consumers to buy LSIS. Across all three types of the intervention materials, a common theme of LSIS use, embedded within the broader strategy to reduce blood pressure through physical activity, healthy diet and regular follow-up with healthcare providers, was communicated. The message, recipient and medium is depicted in Figure 1.

Our study has some strengths. The formative research involved a range of stakeholders who were known to the study investigators through implementation of various research projects in the study sites where they resided. The design of the formative research used a published framework [23] on LSIS use, which helped gain an in-depth understanding of issues around awareness, availability, affordability and taste that were crucial for intervention design. We also acknowledge some limitations. We were unable to recruit salt manufacturers despite repeated efforts. They would have provided information on LSIS production and supply, which is crucial for the implementation of the intervention. Given low awareness of LSIS, including them in the research would not have provided insights into awareness regarding LSIS, which was found to be very low. Similarly, we were unable to recruit retailers in one site to participate in focus groups, which may have provided some additional insights into LSIS supply and sale specific to that site.

## 5. Conclusions

This formative research showed low awareness and availability of LSIS in the study sites. To improve the uptake of LSIS, an intervention to improve awareness, and thereby availability, through trained project staff using various educational materials was developed and implemented.

## Figures and Tables

**Figure 1 nutrients-16-00130-f001:**
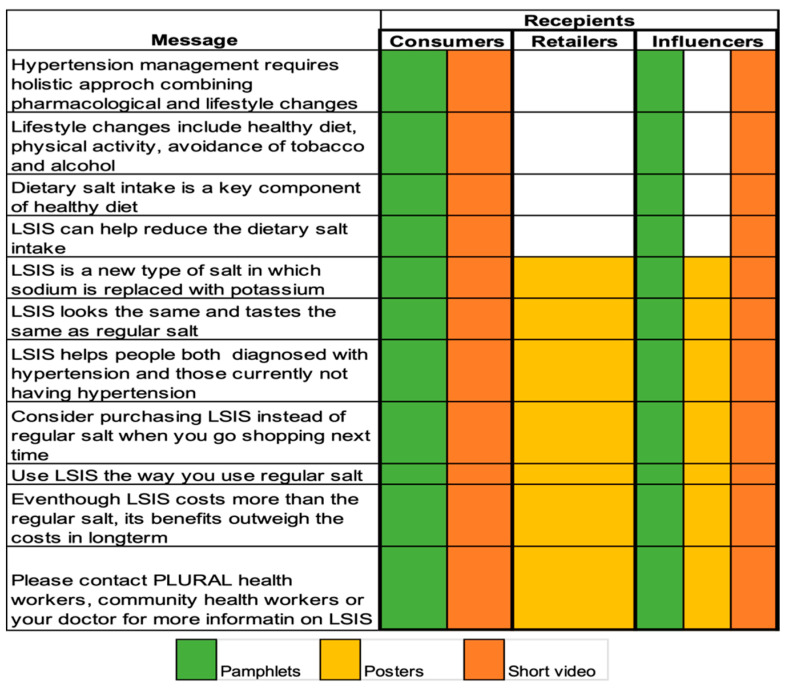
Messages, recipients and the medium of dissemination of the PLURAL intervention messages.

**Table 1 nutrients-16-00130-t001:** Topics explored during in-depth interviews and focus group discussions with study participants.

Participants	In-Depth Interview Topics Individual Knowledge and Experience Related to	Focus Group Topics Community Perspectives on
Consumers	1. Blood pressure, hypertension and salt intake.	1. LSIS facilitators and inhibitors (general concerns, cost, incentives, potential of subsidy). 2. Intervention delivery (health campaigns, trusted sources and mode of intervention).
	2. Diet and lifestyle changes to reduce hypertension.	
	3. Salt consumption behavior	
	4. Salt reduction (importance, challenges).	
	5. LSIS perception (general awareness, reservations about quality, taste, texture, safety and cost).	
Retailers	1. Hypertension and salt intake.	1. LSIS facilitators and inhibitors (supply chain concerns, incentives, potential of subsidy). 2. Intervention delivery (health campaigns, trusted sources and mode of intervention delivery and role of retailer).
	2. Diet and lifestyle to reduce hypertension.	
	3. Salt purchase behavior and preferences of consumers.	
	4. Salt reduction.	
	5. LSIS perception (general awareness knowledge, availability, cost concerns, demand and supply).	
Influencers	1. Patients diagnosed with hypertension and CVDs.	
	2. Salt reduction by patients.	
	3. LSIS perception (potential concerns).	
	4. Strategies for promotion of LSIS.	

CVDs—Cardiovascular diseases; LSIS—Low Sodium Iodized Salt.

**Table 2 nutrients-16-00130-t002:** Total number of interviews and focus groups conducted.

	Visakhapatnam and Anakapalli			Sonipat			Total
	Consumers	Retailers	Influencers	Consumers	Retailers	Influencers	
In-depth interviews	12	12	8	8	14	8	62
Focus groups	2	2	0	2	0	0	06

**Table 3 nutrients-16-00130-t003:** Summary of findings from in-depth interviews and focus groups.

Participant Type	Key Findings
Consumers	1. Participants were able to list either the symptoms or risk factors of hypertension and complications of uncontrolled hypertension.
	2. Taste was mentioned as an important factor in transitioning to low sodium diet.
	3. Most of the participants were unaware of LSIS.
	4. Cost was reported as a major barrier to purchase LSIS.
	5. Subsidy as an intervention to reduce the price of LSIS was welcomed.
	6. Healthcare professionals were mentioned as trusted source for promoting LSIS.
Retailers	1. Majority of the participants reported high cost and low demand as the reasons for not selling LSIS.
	2. Subsidy to offset higher cost of LSIS was welcomed.
	3. Willingness to participate in the intervention was expressed by all the participants.
	4. Healthcare providers were reported as the trusted source of information. Involvement of other stakeholders was also encouraged.
Influencers	1. Participants reported difficulty in adhering to the low salt diet by patients.
	2. Lack of awareness and availability of LSIS along with high cost was a major barrier to its use.
	3. The need for interpersonal communication to convey the message regarding LSIS as part of the intervention was highlighted.

LSIS—Low Sodium Iodized Salt.

## Data Availability

The data that support the findings of this study are available from the corresponding author upon reasonable request. The data are not publicly available due to ethical reasons.

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
