# Peer review of "Awareness and Availability of Low Sodium Iodized Salt: Results from Formative Research of Promoting Uptake of Low SodiUm Iodized Salt by Rural and Urban HousehoLds in India—The PLURAL Study"

_nutrients, 2023, doi:10.3390/nu16010130_

Round 1

Reviewer 1 Report

Comments and Suggestions for Authors

Abstract:

Line 15. Suggest rewording to ‘salt intake is high in Indian adults (?) and is an important…’

Your abstract seems to be missing a conclusion/implications of findings section. I suggest adding a sentence here.

Introduction:

Line 37. Add in ‘dietary’ before sodium.

Line 38. Please state the actual dietary sodium (or salt equivalent) intake in Indian adults and clarify that this is in adults (or children if relevant).

Line 49. Does the discretionary salt use include salt added at the table? Can you provide some detail about this? Is the 80% only that which is added in cooking? Out of the remaining 20% , how much can be attributed to salt added at the table? Perhaps you can provide some commentary around how common adding salt at the table in India is.

Line 57. Change shows to ‘have shown’.

There seems to be an aim missing at the end of your introduction. Is the last paragraph, starting at line 62, proposing that you’re aiming to assess this in your study? Just clarify here.

Materials and methods:

Line 70. Reword ‘the phase 1 of PLURAL study’ to phase 1 of the PLURAL study…

Line 72. Which study sites specifically?

Line 76. What is meant by ‘in-depth’ interviews? Do you mean face-to-face or online interviews?

Line 84. How did you recruit participants from varying socio-economic backgrounds? What approach did you take for this? Provide more details here.

Line 85. What was your rationale for only including participants aged 30 years and over? If this is based on previous research, can you please cite said research/papers?

Line 96-101. You don’t really provide much detail about the participants. Despite being a qualitative study, can you provide some details about the sample size and how many participants you were aiming to recruit and how you did this?

Line 102. You have provided superficial details about the interview and focus groups here. Can you elaborate on the following: Interview length? Where were the interviews and focus groups conducted? Who conducted the interviews? Were the interviews conducted face-to-face or online or over the phone? How were participants contacted for the interviews and focus groups and how did they give consent? Can you also provide an interview guide (as an appendix) detailing how many questions were included and how these questions were selected to facilitate discussion?  How many focus groups were conducted and how did you create homogenous groups among these? Can you provide more details about reaching saturation point in the focus groups? How was the information from the focus groups documented and was a moderator/facilitator involved in the discussion?

Line 120-1. Given that you state that you translated the interviews and focus group discussions to English, can you clarify which language/s these were conducted in?

Line 121. Include the version no. for NVivo.

Line 123. Have you defined RS earlier?

Results:

Line 131. If you’re going to present some details about the participants characteristics, can you include the % of males vs. females in the study and perhaps the age range (with the mean age which you have already reported)? Do you have the details about the number of participants from each site? You state that you provide this in table 2 but it is not in fact presented.  

Line 132. Why were there no participants in the influencers focus group?

Line 147. Use the terms ‘female’ or ‘male’ instead of woman or man in direct quotes you have presented throughout the results section.

Discussion:

Line 260. At the start of the discussion can you just highlight the aim of the study and then discuss the main findings? Revise ‘from the formative research’ to ‘from the current study’.

Line 272. The knowledge of the link between salt intake and hypertension is not a surprising result with other studies showing that this is common among most people. Do you have other data that you can site here from the PLURAL study to show that awareness has in fact increased as a results of these messages or research projects and the reach to the public?

Line 276. Participants are unaware of the effects of LSIS on health because they are generally unaware of LSIS altogether. What is the key point you’re trying to make here- can you please clarify?

Lines 284-5. ‘given the 284 low overall awareness and availability of LSIS at study sites. Because of the low awareness, participants did not report any concerns around the safety of LSIS’. These sentences do not make sense. Can you revise and elaborate on what you mean here?

Line 288. How has the PLURAL study increased awareness of the LSIS? Have these findings been published?

Lines 292-3. Do you have any data on the reach of the PLURAL intervention? More details needed here.

Line 307. Can you cite the frameworks you mention here? Also, at the start of this section, you state that the study has many strengths and then state only two. Revise this sentence or state more strengths of the study.  

Lines 313-315. You also were unable to recruit any influencers for the focus groups, but you only make mention of the retailers of whom you did recruit 2. Can you elaborate on why you think you were unable to recruit any influencers for the focus group?

Comments on the Quality of English Language

This manuscript is mostly well written however, there are a number of grammatical errors throughout the manuscript. Can you please ensure that you carefully read the next draft and amend these errors. 

Reviewer 2 Report

Comments and Suggestions for Authors

Rarely I read such an interesting and well-written paper under review.

English language is fine, I found no mistakes in the text.

I am perfectly fine with the conclusions of the authors.

Nevertheless few minor concerns arise.

First the less significant: in the introduction (row 42) an abbreviation is cited for the first time in the text without its meaning (I suppose Cardiovascular diseases...)

Second, my major concern: in the method section I cannot find the number of subjects under scrutiny. My hope is that the total number studied is not the number of the interviews administered (62) as reported in the result section.

Reviewer 3 Report

Comments and Suggestions for Authors

In my opinion, this study is important to the Indian people and other populations looking for alternatives to reduce salt consumption to prevent hypertension. It is a qualitative study aimed at evaluating the contextual understanding of the issues around the topics of hypertension and its risk factors; the relationship between dietary salt and hypertension; salt purchase and consumption pattern; low sodium iodized salt (LSIS) availability, affordability, taste and safety among consumers, retailers, and influencers of rural and urban households in the North (Sonipat district) and South (Visakhapatnam and Anakapalli districts) in India. However, the objective was not explicit in the abstract or at the end of the introduction. I could not analyze if the conclusion was adequate because of this.

The references are up to date, 16 of 24 from the last five years.

Comments on the Quality of English Language

It is necessary to review the English writing.

Round 2

Reviewer 2 Report

Comments and Suggestions for Authors

I believe the number of subjects studied should be increased a lot before a decision about a possible publication may be reached.

Author Response

Pleas

We thank the reviewer for the comments and helpful suggestions. We provide the response below to their comment.

  1. I believe the number of subjects studied should be increased a lot before a decision about a possible publication may be reached.

Response: We have interviewed 62 and 60 participants for in-depth interviews and focus groups respectively. Even though we had initially planned to conduct 40 in-depth interviews and four focus groups, we ended up doing  62 interviews and six focus groups. The decision to increase the sample size was guided by data saturation. We believe the number of interviews and focus groups is adequate and at this stage, we are unable to increase the number of subjects.
